# Transformational leaders' approach to overcapacity: A study in correctional institutions

Heni Yuwono[1], Desynta Rahmawati Gunawan[2], Anis Eliyana[2]*, Rachmawati Dewi Anggraini[3], Pandhu Herlambang[2], Nurul Iman Abdul Jalil[4]

1 Directorate General of Corrections, Ministry of Law and Human Rights of the Republic of Indonesia, Central Jakarta, DKI Jakarta, Indonesia, 2 Department Management, Universitas Airlangga, Surabaya, East Java, Indonesia, 3 Research and Publication, PT Usaha Mulia Digital Indonesia (PT UMDI), South Jakarta, DKI Jakarta, Indonesia, 4 Department of Psychology and Counseling, Universiti Tunku Abdul Rahman, Kampar, Perak Darul Ridzuan, Malaysia

* anis.eliyana@feb.unair.ac.id

**Data Availability Statement:** All files are available from the Mendeley database: https://data.mendeley.com/datasets/dsg4f49xny.

## Abstract

This research attempts to examine the effect of transformational leadership (TL) on job satisfaction (JS) and job performance (JP) mediated by leader-member exchange (LMX) at all correctional officers in West Java. The quantitative method is adopted by conducting a questionnaire survey on all officers in the West Java area, with a total of 215 respondents. The questionnaire was done through the Google Forms platform and distributed in a time-lagged method. The sample of respondents who were interviewed was obtained through a purposive sampling technique, namely taking samples with certain considerations. The data in this study was then analyzed using Structural Equation Modeling (SEM), which was assisted by the AMOS program. The findings revealed that TL has a large impact on JS, JP, and LMX, whereas LMX has a significant impact on JS and JP. This study contributes to the literature by linking TL, LMX, JS, and JP. It examines and analyzes how and why TL and LMX can affect JS and JP.

## Introduction

Human resource management's role is to arrange employees so that they can work and perform at their best, as well as to make them feel at ease at work [1] since human resources are considered one of an organization's most valuable assets [2]. During 2020–2021, the continuing COVID-19 epidemic wreaked havoc on economies and lifestyles all around the world, specifically in Indonesia [1]. Because of the numerous policy adjustments that had to be undertaken, this pandemic caused the whole organization additional problems in executing their duties and attaining organizational goals.

A common problem in correctional institutions is that the population has been increasing since the early 1900s, in which more than 9 million people have been imprisoned worldwide [3]. Meanwhile, in Indonesia, data from the Directorate General of Corrections revealed that the number of prisoners in 2020 reached 249,139 people. This number far exceeded the capacity of only 135,561 people. In other words, there was an overcapacity of 84%. Other situational hurdles included fights between prisoners, destruction of public facilities, and internal

**Funding:** The author(s) received no specific funding for this work.

**Competing interests:** The authors have declared that no competing interests exist.

problems of prisoners. Therefore, working in correctional units possesses very high risk, and causes work pressure [4].

Good performance can be achieved if organization leaders can manage human resources [5] because employee performance is also influenced by the role and behavior of the managers themselves [6]. As the complexity of the organizational environment increased in all aspects during a pandemic, there was a need for job adjustments and an increase in the effectiveness of the relationship between leaders and subordinates [7]. Finding the right leadership style is considered essential, and TL is one of the modern concepts in management [8]. Today's organizations need leaders who can form a clear vision of the future, plan strategies for developing and managing change, and avoid crises, to keep up with modern developments and trends in information and communication technology for the sake of the organization [9].

When Indonesia faced the global Covid-19 crisis, TL could be a useful resource to continue organizational activities by facilitating employees and maintaining strategies that were able to create increased employee performance [10, 11]. To be specific, TL is considered as an approach because leadership is an organizational approach to undergo a time of crisis. When communication is hampered, and work is disrupted as it is today, TL will still be able to affect employee JS [12]. According to Locke, referenced by [13], JS is a pleasant emotional state arising from an evaluation of one's job as a successful accomplishment within the context and substance of the employment. Following its nature, TL is very strategic, aiding personnel at all levels and assisting them in overcoming all conceivable hurdles to meet current problems [12]. This study evaluates the influence of the appropriate TL style on employees' JS, taking into account the existing organizational issues during Covid-19 and the prospective nature of TL [14].

Much research back up the notion that the relationship between employees and their leaders might influence JP [15]. This supervisor-subordinate connection is commonly referred to as a leader-member exchange in the literature (LMX). The construction of work conceptions that center around various components of work assignments and positions is a part of the LMX [16]. Leader-member exchange is also expressed as a measure of the quality of the relationship between leaders and subordinates, including loyalty, understanding, trust, and expertise [17]. This definition primarily evaluates relationships from the perspective of subordinates, but recent research has also examined leader views and the binary interaction between leader and follower views [15]. A well-developed LMX concept will improve employee JP and help leaders to achieve goals more effectively.

TL theory and leader-member exchange will be influential in the business and management literature [17] because they explain how leadership affects team performance and employee JS. Based on the results of interviews with the Secretary of the Directorate General of Corrections from the Indonesian Ministry of Law and Human Rights during the pre-research, the empirical phenomena of JP and JS which are influenced by TL and LMX have not been widely carried out in the context of correctional institutions. In the context of this study, appropriate leadership will make correctional officers able to maintain the quality of work despite all the challenges that occur during the pandemic. Specifically, this research conducted in all correctional units in West Java will examine the effect of TL mediated by LMX on JS and JP. This study is unique compared to previous studies because it has two affected variables.

## Literature review

### Theoretical foundation

**Transformational leadership.** Leadership is a concept of power that has the potential to influence and help groups of individuals achieve goals [18]. Transformational leadership is

part of the "New Leadership" paradigm, which prioritizes charismatic and affective elements, and acts as a change agent to create and articulate a clear vision for an organization [18]. Transformational leadership can empower followers to meet higher standards by making others want to trust them, and giving meaning to organizational life well. TL is the most fundamental determinant of the success or failure of a transformation attempt. TL is defined as a leadership style in which leaders and followers collaborate to improve each other's motivation and morale [18]. TL tends to be open-minded and visionary, which can be a motivation for employees to work beyond expectations [19]. Alhashedi et al. describe that TL tends to influence employees more optimistically by encouraging and initiating organizational change, carrying out changes only with the help of other transformation agents, and these agents work with different competencies and skills that are changed based on circumstances and requirements [7]. TL is known to be an effective leadership style in encouraging behavior that can direct changes in organizational structure, strategy, mission, and culture to promote the product and work innovation [20]. Inspirational motivation, intellectual stimulation, individual consideration, and idealized influence are some of the influence methods used by transformational leaders [21].

**Leader-member exchange.**   Leadership may be defined as the process of an individual influencing a group of others toward a common goal, which includes overcoming obstacles encountered by both leaders and their followers [18]. In other words, the leader and follower constitute an inseparable unit. The majority of leadership theory discussions in the literature focus on leadership only from the leader's perspective (e.g. trait approach, skills approach, and style approach) or vice versa from followers and context (Situational Leadership and path-goal theory) [18]. In contrast, the leader-member exchange (LMX) paradigm conceptualizes leadership as a process that revolves around the interaction between leaders and followers. Leader-member exchange theory refers to the dyadic relationship between leaders and followers, and how social exchange creates and maintains the quality of those relationships [17]. As a result, two groups of followers emerge: one is the inner group, and the other is the outside group [21]. This is due to the nature of the relationship. The in-group has a 'high-quality exchange relationship' with the leader, whereas the out-group has a 'more formal' relationship. Low leader-member exchange connections are more commercial and are marked by formal agreement and a revenge mindset, whereas high leader-member exchange relationships are more social and are marked by support, cooperation, and dedication [17]. Leader-member exchange is also expressed as a measure of the quality of the relationship between leaders and subordinates, including loyalty, understanding, trust, and expertise. According to [22], the notion of leader-member exchange can be used to evaluate management practices that illustrate how members develop leadership networks within the organization to help them carry out their responsibilities more successfully. The leader-member exchange paradigm is also used in a variety of organizational structures, including educational, traditional, and government institutions.

**Job satisfaction.**   JS is defined as a feeling of self-satisfaction and pleasure brought on by an individual's evaluation of their work and experience, which includes their emotions and attitudes toward work [23]. According to Locke cited by [13], it is a pleasant emotional state resulting from an appraisal of one's job as an achievement of individual job evaluations in the context and content of the job. [24] also stated that JS is a common behavior towards work performance when there are appropriate rewards and achievements. JS can also be interpreted as giving quality to professional tasks and describing the satisfaction obtained from the performance of professional roles, as well as the attention given to emotional and cognitive aspects [25]. In general, JS is defined as a person's emotional response, verbal expression, and cognitive evaluation of their job [26]. Thus, JS is expressed as a form of employees' perceptions of how well their work is delivering important things.

**Job performance.**   The achievement of a goal depends on the performance of the organization, namely the ability of an organization to implement strategies and manage its goals effectively [18]. Performance is the stage of achievement of completion of work that represents the level of achievement of each job in completing a particular job [27]. JP can be a construction that is associated with an individual's ability to achieve goals in the hope of meeting job targets, the environment, and the standards set by the organization [28]. JP is the actual result or output of an organization that is measured against the organization's intended output and refers to the achievement of organizational goals and objectives [29]. JP can also be interpreted as a measure of good quality and quantity results within a certain period which is influenced by many factors and is carried out with responsibilities carried out by employees to achieve organizational goals [30]. Ultimately, JP will indicate the degree to which individuals complete task-related behaviors and is an indicator of how well they perform in their job domain [31].

## Hypothesis development

**Transformational leadership on job satisfaction.**   Leadership is the embodiment of a leader's behavior in leading and is declared effective when it can encourage motivation for productivity, organization, loyalty, and JS or increased organizational members [1]. According to this research, TL is the ability to motivate and inspire followers to achieve greater results than planned. TL will be able to communicate an attractive vision and mission for the future of the organization, and if leaders and employees communicate effectively, they can understand each other's needs and demands [12]. TL can share opinions and contribute to the decision-making process and also convey needs to senior management [12]. Thus, employees will express their JS with their work and work environment. Furthermore, TL can also create a positive influence by motivating employees to act toward achieving organizational goals and showing special attention to their employees [32]. A positive association between TL and JS has been discovered in several recent research [1, 12, 33]. The practice of TL needs to be maintained because it can encourage and build harmonious relationships with employees as a driver of JS.

*H1*: *Transformational Leadership Has a Significant Effect on Job Satisfaction.*

**Transformational leadership on job performance.**   Employees will be inspired, encouraged, and motivated by transformational leaders to innovate and make changes that will help the business grow and influence its future success [18]. This leadership focuses on the feelings of employees and encourages their employees to do their best in the workplace by being able to create JP. TL can help build JP above individual aspirations, by inspiring development and contributing to organizational performance [8]. This approach explains that TL strongly influences organizational outcomes, and several researchers have suggested that the idealized influence of TL is most likely to change organizational outcomes, including JP outcomes. Transformational leaders influence JP through inspiring and stimulating expectations, and by providing a commitment to organizational goals [34]. Meanwhile, TL will also suggest outcomes, such as creativity, innovative behavior, product development, and JP [35]. [36] explained that leaders tend to improve their company's JP by providing an understanding of the company's mission and being role models. TL is also known to be able to predict future possibilities and map substitution strategies to meet uncertainty [37]. Research from [7, 8, 18, 38] stated that there is a positive relationship between TL and JP.

*H2*: *Transformational Leadership Has a Significant Influence on Job Performance.*

**Transformational leadership on leader-member exchange.** Without being physically there, team leaders must be able to establish trust and connection with team members [39]. A high-quality relationship with a leader, i.e., an LMX, will be characterized by increased responsibility, decision influence, and easy accessibility to resources [40]. TL, which focuses on reaching beyond work goals, goals, and high-level intrinsic requirements, is typically considered a leadership style that is beneficial for teams, among other leadership styles [41]. TL motivates individuals to take on specific activities by recognizing each employee's unique talents, stimulating new ways of thinking and solving challenges, and establishing a common vision and goals among employees [17]. Based on evidence that the two dimensions complement and impact one other, experts have argued for a more integrated study of TL and LMX. For example, one study suggests that transformational leaders maintain high-quality member exchange leaders because employees are more receptive to interaction because of their charismatic appeal [42]. Furthermore, through individual exchanges that establish LMXs, TL can be 'personalized' [43]. Other studies with similar results include [17, 21, 40, 44].

*H3*: *Transformational Leadership Has a Significant Effect on Leader-Member Exchange.*

**Leader-member exchange on job satisfaction.** Employees experience many interactions and social relationships with colleagues, managers, subordinates, and clients during work, however, the relationship with their immediate supervisor is the most important [45]. The relationship formed by the LMX will create JS because it can form good relationships that will later help work. High-quality relationships will lead to a positive atmosphere and the occurrence of JS at a higher level [22]. According to [46] Employees that participate in high-quality leader-member exchange interactions are more likely to display positive improvements, which leads to JS. The concept of LMX is derived from social exchange theory and role theory [47]. This demonstrates that high levels of LMX often result in superiors and subordinates having a high level of understanding, trust, and commitment [48]. In addition [45, 48], found that the higher the LMX, the higher the employee JS. When subordinates are more satisfied with their leader, their work stress and job discomfort are much reduced [49].

*H4*: *Leader-Member Exchange has a Significant Effect on Job Satisfaction.*

**Leader-member exchange on job performance.** High LMX relationships will be more social and characterized by support, reciprocity, and commitment [17]. LMX is also a measure of the quality of the relationship between leaders and subordinates, including understanding, loyalty, trust, and abilities that help employees improve their JP. The degree of reciprocity of the leader with followers with other resources such as information and opportunities to participate in the decision-making process is affected by the quality of the LMX, and high-quality LMX is distinguished by higher obligations concerning follower performance as a result of leaders' commitment [50]. According to [51] there is a relationship between leader personality traits and JP that has been studied where leaders can control tasks and identify the JP of their followers. They considered the impact of similarity and contrast in the proactive personality traits of a leader and follower pairs on followers' JP. Individuals can use social categorization to understand their own identity, the identity of others in their environment, and the identification of members within and outside the group through interactions with other ingroup members. They are then motivated to engage in behaviors that reinforce membership in a valuable identity group through interactions with other ingroup members. [16] which is important to produce the right performance. Supported by [16, 51, 52] stated that there is a positive relationship between LMX and JP.

*H5*: *Leader-Member Exchange has a Significant Effect on Job Performance.*

**Leader-member exchange mediates the effect of transformational leadership on job satisfaction.** JS will be related to organizational variables that are influenced by organizational culture and leadership style. This also helps increase productivity and efficiency which has an impact on JS. JS has been used as a means to retain qualified employees in the organization, and this requires the influence of TL to make it happen [53]. That is, being a team member at work needs to have a high-quality LMX relationship because it is considered a profitable and ideal position in the organization [54]. TL can influence the positive work attitudes of employees by creating more relationships within the group with JS. One of the contributions of this research is its ability to demonstrate that TL that affects employee satisfaction achieves this through building high-quality relationships with their subordinates (LMX) [54]. This is of practical importance because highly competitive and creative organizations rely on high-ranking JS as a competitive advantage. High levels of mutual trust, interaction, and support, as well as a high level of reciprocity in which both parties give to resources that are appreciated by the other, are all hallmarks of a high-quality LMX relationship that can lead to JS [41]. According to [54], the leader-member interchange can mediate the correct leadership style-JS relationship.

*H6*: *Leader-Member Exchange Significantly Mediates the Effect of Transformational Leadership on Job Satisfaction.*

**Leader-member exchange mediates the effect of transformational leadership on job performance.** According to the LMX theory, leaders will influence employees to have high-quality relationships with their leaders, which will allow them to obtain a greater share of the leader's attention, resources, and support than second-hand employees with lower-quality ties [55]. Furthermore, it will trigger positive things by recommending building effective communication in all organizations to strengthen organizational loyalty and lead to better JP [12]. Because of its strong and persistent link with high-JP teams, TL has become a key subject of interest in organizational psychology and has garnered a large quantity of research [56]. Rooted in social exchange theory, LMX concentrates on dyadic exchanges between leaders and followers in addition to the process of relationship development showing that LMX incorporates operationalization of a relationship-based approach to leadership based on the benefits of both parties [55]. Such high-quality relationships are desirable because LMX was found to predict JP in various settings and conditions [44] which can also be passed through TL which is often considered the most effective leadership style in building team morale and performance. high followers [21]. Supported by [55] which states that the LMX can mediate the right leadership theory relationship in influencing JP.

*H7*: *Leader-Member Exchange Significantly Mediates the Effect of Transformational Leadership on Job Performance.*

The following Fig 1 summarizes the overall hypotheses of this study.

## Method

### Data collection procedure

This study is based on a predetermined research model and data obtained only from research samples that meet the requirements of 215 respondents. The questionnaire was done through Google Forms, with a time-lagged method because the questionnaire was distributed twice.

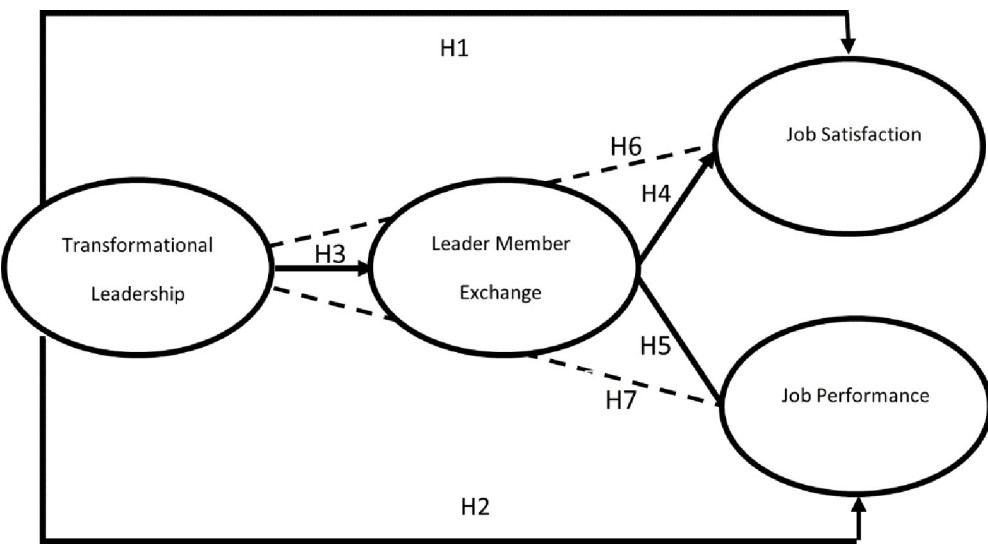

**Fig 1. Research method.**

Google Form links are disseminated through social media such as telegram and/or Whatsapp. This research has fulfilled the licensing requirements from the head manager and is carried out outside office hours so that it does not interfere with the work activities of employees. The subjects of this research are officers who work in correctional units. The sample used is all correctional officers in the West Java area. With the use of LMX and TL, this study aims to improve the performance and satisfaction of correctional officers. This study used a quantitative method to achieve its aim. The research design includes procedures for data collection, measurement, and data analysis.

As this is a non-interventional study, the Research Ethics Committee of Universitas Airlangga: Development and Innovation Institute of Publishing Journal and Intellectual Property Rights (LIPJIPHKI) determined that no ethical approval was necessary. In addition, written informed consent was acquired from the head manager, who represented the whole organization's participation, under the organization's policy. The consent was also accepted by the Development and Innovation Institute for Publishing Journal and Intellectual Property Rights (LIPJIPHKI) of Universitas Airlangga.

## Measurement

In testing the hypothesis of this study using a quantitative approach. The independent variables used in this study are TL (X1) and JS (X2), then the mediating variable used is LMX (Z), and the dependent variable used is JP (Y). This study uses a measurement scale with the Linkert scale to measure the response to the statements contained in the questionnaire. The Linkert scale used in this study has five scales ranging from 1 (strongly disagree)– 5 (strongly agree). This study measures the TL (X1) variable using items that refer to the journal belonging to [57], then the LMX (Z) variable uses items that refer to [58], the JS variable (Y1) uses items that refer to [59], and for the JP variable (Y2) using items that refer to the journal belonging to [60].

## Data analysis techniques

This study analyzed the data using the multivariate analysis method to test several relationships between variables. This research instrument uses the Structural Equation Modeling (SEM)

method on the Amos v.24 application. SEM is a combination of factor analysis and regression and is expressed as a multivariate analysis that can analyze the relationship between variables in a more complex manner. Furthermore, SEM will be used to determine the magnitude of the influence of the independent variable on the dependent variable.

## Data analysis

Respondents in this study were correctional officers at the Ministry of Law and Human Rights in West Java. The characteristics of respondents were described by gender, length of work, and level of education. The majority of employees are male, amounting to 151 people or 70.2%, the majority with an age level of 31–40 years, amounting to 87 people or 40.5%, the majority of working years > 8 years being the largest number of respondents with a total of 135 people or 62.8%, and the majority with a high school education level / vocational school / the like with a total of 100 people or 46.5%.

All of the respondents' answers on transformational leadership, leader-member exchange, job satisfaction, and job performance variables indicate strongly agree because they have an average in the range > 4.20–5.00. Then, the results of the normality test showing a multivariate c.r of 52.33 which is in the range –1.96 to +1.96 at a significance level of 5%, meaning that the multivariate data are not normally distributed. This condition is not a significant problem because the MLE estimation technique in SEM is efficient and unbiased, both when the multivariate normality assumption is met or not and has been proven to remain robust (robust) in the event of a violation of the normality assumption. Thus, the analysis can proceed to the next stage.

9 observations have a d-squared Mahalanobis value greater than the chi-square limit of table 65.25, thus the 9 observations (respondents) are indicated as outliers and subsequently dropped from the analysis. Thus, the sample size was reduced by 9, from 215 respondents to 206 respondents. The summary of the results of the suitability test of the measurement is presented in the model in Fig 2.

The results of the validity and reliability test show that all the variables have a factor loading value > 0.50, hence these indicators are valid in forming the construct and building the model. In addition, a construct in this study is said to be reliable because it has a construct reliability value > 0.70. Then, the results of the goodness of fit index values produced by the structural model are presented in the model in Fig 3.

The results of the structural model suitability test on all absolute fit indices and incremental fit indices criteria have met the requirements (marginal fit and good fit) 2 = 0.000, Cmin/df = 1.617, GFI = 0.811, RMSEA = 0.055, SRMR = 0.053, CFI = 0.924, TLI = 0.918, AGFI and PNFI are not used because they are only used for comparisons between two alternative models, so that the structural model is acceptable, and then tested the significance of the influence between variables, both direct and indirect effects direct.

Based on Table 1, it is known that the significant effect between variables if the CR value is 1.96 or p-value 5% significance level, then it is decided that there is a significant effect between these variables.

Based on Table 2, it is known that if the significant influence between variables uses the provisions if the CR value is 1.96 or the p-value is 5% significance level, it is decided that there is a significant effect between these variables.

The results of the comparative analysis of the mean and total effect variables (Fig 4) conclude that to improve JS and JP for correctional officers within the Ministry of Law and Human Rights, West Java, the priority is TL, then leader-member exchange.

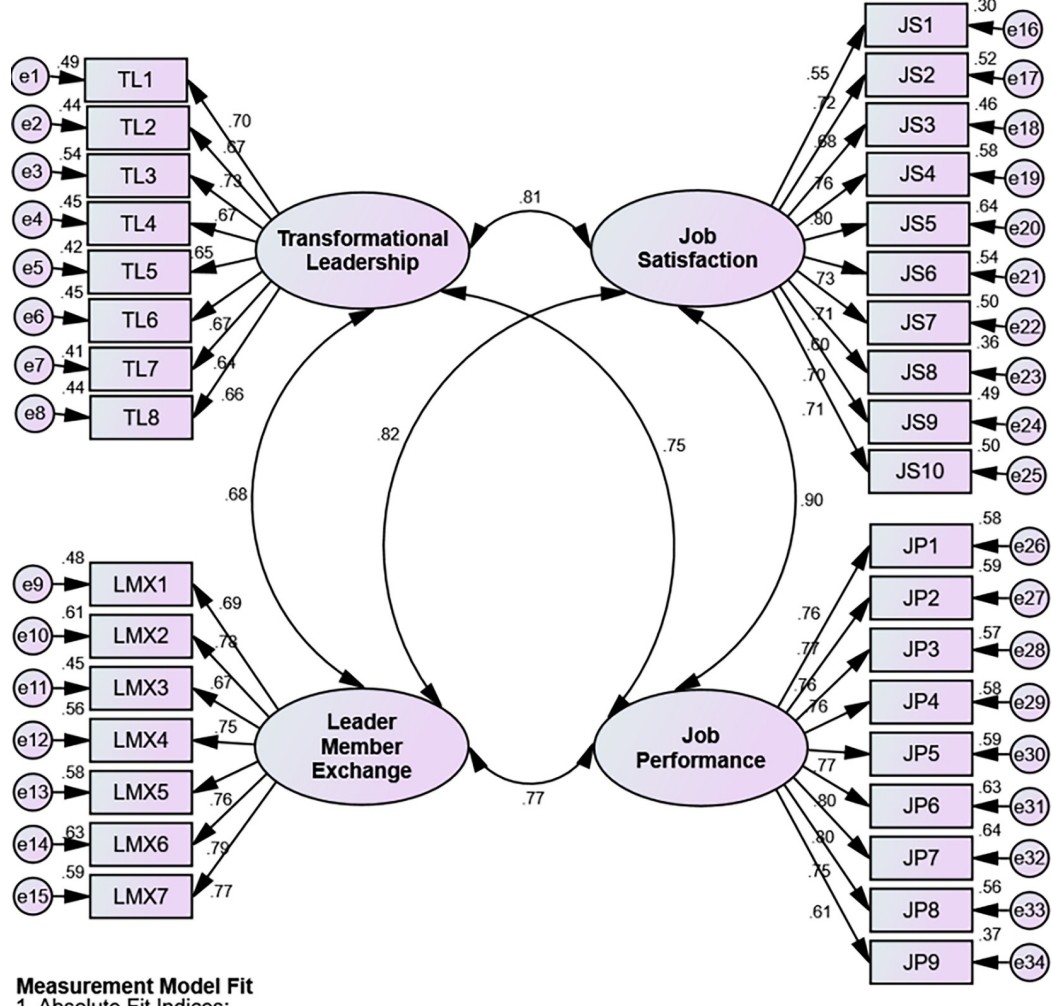

**Measurement Model Fit**
1. Absolute Fit Indices:
   Chi-Square = 826.362     Probability = .000     Cmin/DF = 1.586     GFI = .818     RMSEA = .052     SRMR = .048
2. Incremental Fit Indices: CFI = .928     TLI = .923     NFI = .828     RFI = .815
3. Parsimony Fit Indices: AGFI =.793     PNFI =.769

**Fig 2. Assessing the measurement model.**

## Results and discussion

### Discussion

Based on data processing using the Structural Equation Modeling (SEM) method on the Amos application, it was found that the parameter estimation result shows that TL had a significant effect on JS with a CR value of 5.859 ($>$ 1.96), a significance value (p-value) of 0.000 ($<$ 5% significance level), and the effect coefficient is 0.538 (positive). Thus, H1 is accepted. This result is in line with previous research from [1, 12, 33]. The results show that TL can communicate an attractive vision and mission for the future of the organization, and correctional leaders and officers at the Ministry of Law and Human Rights in West Java can communicate effectively which allows them to understand each other's needs and demands, especially in a communication that is also is a major determinant of mutual understanding and JS.

The parameter estimation result shows that TL has a significant effect on JP with a CR value of 5.657 ($>$ 1.96), a significance value (p-value) of 0.000 ($<$ 5% significance level), and the

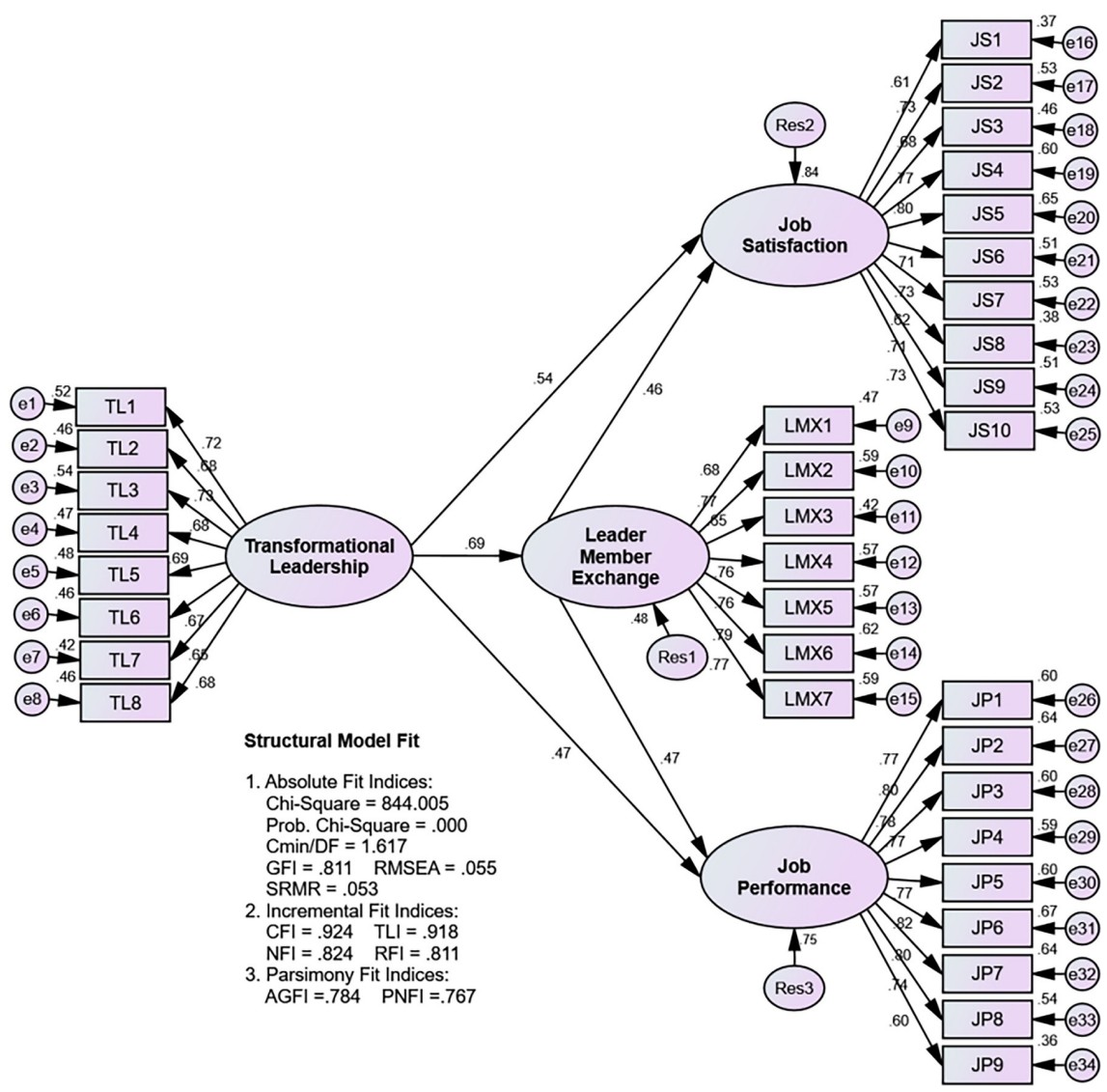

**Fig 3. Assessing the structural model.**

**Table 1. Fit measure for the structural model.**

| Hip. | Structural relationship | | | Std. Estimate | C.R. | P value |
|------|------------------------|---|---|---------------|------|---------|
| $H_1$ | Transformational Leadership | → | Job Satisfaction | 0.538 | 5.859 | 0.000** |
| $H_2$ | Transformational Leadership | → | Job Performance | 0.472 | 5.657 | 0.000** |
| $H_3$ | Transformational Leadership | → | Leader-Member Exchange | 0.690 | 7.459 | 0.000** |
| $H_4$ | Leader-Member Exchange | → | Job Satisfaction | 0.456 | 5.293 | 0.000** |
| $H_5$ | Leader-Member Exchange | → | Job Performance | 0.471 | 5.500 | 0.000** |

Note

*. Significant at the 0.05 level

**. Significant at the 0.01 level

**Table 2. Fit measure for the structural model.**

| Hip. | Structural relationship | Std. Estimate | Lower bounds | Upper bounds | P-value | Type of mediator |
|---|---|---|---|---|---|---|
| $H_6$ | Transformational Leadership → Leader-Member Exchange → Job Satisfaction | 0.314 | 0.041 | 0.519 | 0.025* | Partially mediation |
| $H_7$ | Transformational Leadership → Leader-Member Exchange → Job Performance | 0.325 | 0.080 | 0.523 | 0.015* | Partially mediation |

Note

*. Significant at the 0.05 level

**. Significant at the 0.01 level

resulting coefficient of influence is 0.472 (positive). Thus, the higher the TL, the higher the JP of correctional officers at the Ministry of Law and Human Rights in West Java. Thus, H2 is accepted. This result is supported by [7, 8, 18]. In addition, this result is in line with a previous study that TL can show a significant improvement in performance by being able to show appreciation for good efforts based on TL's skills [61]. The results show that TL can help build JP above the aspirations of correctional officers at the Ministry of Law and Human Rights in West Java, by inspiring development and contributing to organizational performance. This approach explains that TL greatly influences organizational outcomes, including JP results.

Next, the parameter result shows that TL has a significant effect on the leader-member exchange with a CR value of 7,479 ($> 1.96$), a significance value (p-value) of 0.000 ($< 5\%$ significance level), and the resulting coefficient of influence is equal to 0.690 (positive). Thus, H3 is accepted. In line with this result are studies from [17, 21, 40, 44]. The results show that TL

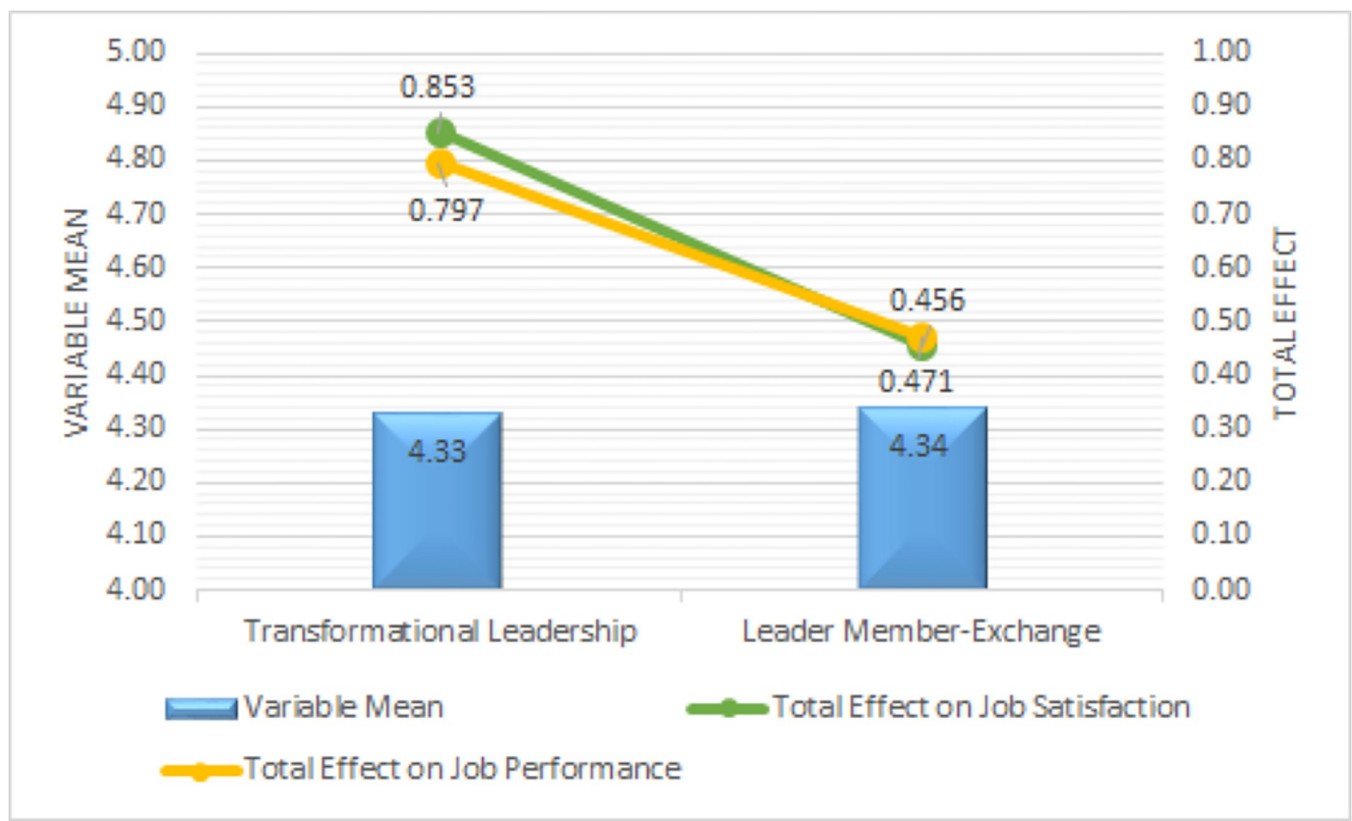

**Fig 4. Comparison between variable mean and total effect.**

inspires correctional officers at the Ministry of Law and Human Rights in West Java to take certain actions by considering their unique individual talents, stimulating new ways of thinking and solving problems, and creating a shared vision and goals among correctional officers. This will demonstrate the integration of TL and LMX based on evidence that the two dimensions of leadership complement and influence each other with correctional officers. TL at the Ministry of Law and Human Rights in West Java can be 'personalized' through individual exchanges that build LMXs.

The parameter result shows that the leader-member exchange has a significant effect on JS with a CR value of 5.293 ($> 1.96$), a significance value (p-value) of 0.000 ($< 5\%$ significance level), and the effect coefficient is equal to 0.456 (positive). So that the higher the leader-member exchange, the higher the JS of correctional officers at the Ministry of Law and Human Rights in West Java. Thus, H4 is accepted. Some previous studies [45, 48, 62] supported the significant influence of LMX on JS. The results of the study show that the relationship formed by the LMX will also create JS for correctional officers at the Ministry of Law and Human Rights in West Java because they are considered able to form good relationships that help work. These high-quality relationships lead to the emergence of positive tensions and lead higher levels of JS for correctional officers at the Ministry of Law and Human Rights in West Java.

The following result shows that the leader-member exchange has a significant effect on JP with a CR value of 5,500 ($> 1.96$), a significance value (p-value) of 0.000 ($< 5\%$ significance level), and the resulting coefficient of influence is equal to 0.471 (positive). So that the higher the leader-member exchange, the higher the JP of correctional officers at the Ministry of Law and Human Rights in West Java. Thus, H5 is accepted. Supported by [16, 51, 52] stated that there is a positive relationship between LMX and JP. The result shows that the quality of the LMX will affect the level of reciprocity between the leader and correctional officers at the Ministry of Law and Human Rights of West Java with other resources such as information and opportunities to be involved in the decision-making process, and high-quality LMXs are distinguished by higher requirements concerning follower performance, as a result of the commitment of leaders at the West Java Ministry of Law and Human Rights.

The parameter estimation result shows that the TL→LMX→JS indirect path significance test has a significant effect with a significance value (p-value) of 0.025 ($< 5\%$ significance level). Thus, H6 is accepted. The nature of the mediator is known to be partially mediation, meaning that increasing JS can only be done by increasing TL, but if it is also accompanied by strengthening leader-member exchange, the JS of correctional officers at the Ministry of Law and Human Rights in West Java can increase even more. This result is supported by previous research by [54] which states that LMX can mediate the right leadership style relationship in influencing JS. The results of the study show that a high-quality LMX relationship will include characteristics such as a high level of mutual trust, interaction, and support, as well as a high level of reciprocity in which both parties contribute and a resource that is valued by both parties. other. Furthermore, TL at the Ministry of Law and Human Rights in West Java can influence the positive work attitude of correctional officers by creating more relationships within the group with JS.

The parameter estimation result shows that the TL→LMX→JP indirect path significance test has a significant effect with a significance value (p-value) of 0.015 ($< 5\%$ significance level). Thus, H7 is accepted. The nature of the mediator is known to be partial mediation, meaning that increasing JP can only be done by increasing TL, but if it is also accompanied by strengthening leader-member exchange, the JP of correctional officers can be even higher. This is supported by previous research [55], which stated that the LMX can mediate the right leadership theory relationship in influencing JP. The result shows that LMX satisfaction can

combine the operationalization of a relationship-based approach to leadership based on the benefits of both parties because LMX was found to predict JP in various settings and conditions that could be pursued with TL at the Ministry of Law and Human Rights in West Java which is often considered the most effective leadership style in building team morale and high follower performance.

## Conclusions and suggestions

### Conclusion

Based on the results of data processing and discussion of the effect of TL mediated by LMX on the effect of JS and JP, the TL has a significant effect on JS, on JP, on the leader-member exchange, while leader-member exchange has a significant effect on JS and JP. The indirect path of TL→LMX→JS and TL→LMX→JP have a significant effect. The right leadership model will be able to develop good relationships between leaders and followers and help the officers at the Ministry of Law and Human Rights of West Java carry out their duties effectively. The results of the study show that the leadership style with TL and the LMX theory can create JS and JP for correctional officers even during this COVID-19 pandemic. Thus, correctional officers will still be able to create good quality work with all the challenges that occur during the COVID-19 pandemic through the encouragement of good leaders. In addition, this study is contextually novel since it presents empirical information regarding the performance of correctional officials under situations of overcrowding. This gives knowledge to organizations with similar characteristics to Indonesian correctional facilities.

### Theoretical and practical implications

The theoretical contributions of the research draw on leadership theory [18], social exchange [55], and LMX theory [17], in which this study analyzes how TL is associated with LMX to JS and JP. This study has provided evidence of a significant effect of TL on officers' LMX, and TL on JS and JP officers. Leaders who guide and direct officers appropriately will affect good relationships which also have a positive impact on work results. Then with the discovery of a higher quality exchange relationship, this study shows that the leader and officer relationship can facilitate JS and can influence JP for the better. In addition, research supports studies on the indirect relationship that is affected by LMX in the association between TL to JS and JP. These results indicate a partial mediating role. Without LMX, TL can significantly lead to both JS and JP. This study provides the novelty that LMX can significantly mediate the effect of TL on JS and JP. Furthermore, the role of TL to create LMX, JS, and JP appropriately can help organizations be more effective which can be implemented for all correctional officers.

In terms of practical implication, the findings can be used by the organization's management to make recommendations on the impact of TL mediated via LMX on work satisfaction and performance. This would necessitate leaders with the correct leadership style constantly raising awareness and enthusiasm for getting part in problem-solving, especially during challenging times like the current pandemic. As a result, correctional officers will provide positive feedback in the form of positive accomplishments (work satisfaction and high-level JP) and will always endeavor to provide solutions to difficulties to assist leaders in achieving their objectives more efficiently. Furthermore, the relationship between factors may be used to show how to persuade employees to be more optimistic, how to support and initiate organizational transformation, and how to examine management policies that can foster leadership networks across the organization. Additionally, leaders may use the findings to make recommendations on how to help employees complete tasks effectively, create employee perceptions

of how well their work is done, and demonstrate the degree to which employees complete their behavior in producing high-quality work despite all obstacles.

## Limitations and suggestions

The limitation of this study is the distribution of research questionnaires conducted in a cross-sectional manner because studies using this technique can lead to biased results. If the questionnaire is distributed more than once, the real sample responses will be stronger. Therefore, further research is recommended to measure the spread of the questionnaire with multiple distributions to get more accurate and real-world sample answers.

## Author Contributions

**Conceptualization:** Anis Eliyana.

**Data curation:** Desynta Rahmawati Gunawan.

**Formal analysis:** Desynta Rahmawati Gunawan, Anis Eliyana.

**Investigation:** Heni Yuwono, Desynta Rahmawati Gunawan.

**Project administration:** Heni Yuwono.

**Resources:** Heni Yuwono.

**Software:** Nurul Iman Abdul Jalil.

**Supervision:** Nurul Iman Abdul Jalil.

**Validation:** Rachmawati Dewi Anggraini, Nurul Iman Abdul Jalil.

**Visualization:** Rachmawati Dewi Anggraini, Pandhu Herlambang.

**Writing – original draft:** Pandhu Herlambang.

**Writing – review & editing:** Rachmawati Dewi Anggraini.

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
