## [Decision Letter · Decision Letter 0]

16 Aug 2022

PONE-D-22-17395Finding Links Among Transformational Leadership, Job Performance and Job SatisfactionPLOS ONE

Dear Dr. Anis Eliyana,

Thank you for submitting your manuscript to PLOS ONE. After careful consideration, we feel that it has merit but does not fully meet PLOS ONE’s publication criteria as it currently stands. Therefore, we invite you to submit a revised version of the manuscript that addresses the points raised during the review process.

We look forward to receiving your revised manuscript.

Kind regards,

Rogis Baker, Ph.D

Academic Editor

PLOS ONE

Journal Requirements:

4. Please ensure that you refer to Figures 3 and 4 in your text as, if accepted, production will need this reference to link the reader to the figure.

Reviewers' comments:

Reviewer's Responses to Questions

**Comments to the Author**

1. Is the manuscript technically sound, and do the data support the conclusions?

Reviewer #1: Partly

Reviewer #2: Yes

2. Has the statistical analysis been performed appropriately and rigorously? 

Reviewer #1: Yes

Reviewer #2: Yes

3. Have the authors made all data underlying the findings in their manuscript fully available?

Reviewer #1: No

Reviewer #2: Yes

4. Is the manuscript presented in an intelligible fashion and written in standard English?

Reviewer #1: No

Reviewer #2: Yes

5. Review Comments to the Author

Reviewer #1: Thank you for your efforts. Please find the following comments for your consideration to enhance the paper:

Title: Finding what links? The purpose of the study highlighted in the abstract can be used to revise the title for clarity what the study is all about.

Abstract – Purpose, methods, findings and conclusion/implications; no need for some points will be discussed later.

Introduction

• Employee can help… and can prevent… in what sense and are there references to support the claim?

• A common problem in a prison… Authors should be specific and discuss what they are sure about. They also need to provide evidence to support any arguments.

• When Indonesia faces … TL will be a tool or was considered an approach?

• Strong justifications for the study at the end of the introduction section will add value to the study.

Literature Review

An introductory sentence or expressions to set a stage for what is to be discussed in this section will be beneficial.

Transformational Leadership

The study describes or a study describes. Better still, in a study, TL is described as …

[7] describe… should be changed and cited appropriately.

Hypotheses development

H1, H2 and H3 – It’s been established that TL has positive effects on job satisfaction and performance from the literature reviewed and discussed in the preceding sections. So, why the hypotheses again or do you want to explore it as you are not sure it can have positive effects in your context? The same goes for H4 and H5.

Is the figure showing the research methods or framework?

Methods

Can be arranged as follows: research philosophy, approach, instrumentation, sample and sampling technique, data collection procedure, data analysis and ethical considerations

In the data analysis section – Authors should describe the methods employed to analyze the collected data. All the outcomes of the analyses should be presented in the results section. The tables and figures are far too many, though. Perhaps, the authors may want to present critical results helping them to address the goals and hypotheses of the study. Seven hypotheses may be considered too many for a study of this nature.

Discussion should be robustly done using the extant literature and highlighting possible implications.

From the findings and conclusions, it is not clear what the study is contributing to the extant international literature. The authors may want to look at the novelty and unique contributions of the study.

The authors should also look at the in-text citations and fix typos/grammatical errors across the manuscript.

Reviewer #2: The authors of this manuscript have done a good job conducting SEM mediation analysis to assess their hypothesized model, which is appropriate from a logico-deductive standpoint. I think that overall this is an interesting and salient piece that can be appropriate for PLOS One, though I do have some areas of feedback to propose:

1. Writing at times is not as clear as it could be - occasional grammar and spelling issues in particular

2. I do not think the literature review has gone deep enough into core research, especially around the key theoretical pieces from which LMX and other leadership theories derived. I would suggest some additional work in this area - see the Northouse Leadership textbook for some helpful sources.

3. Methodology needs expansion, especially providing some detail on the context/setting (prison) and some additional information/literature support on the methodology

4. Make sure tables and figures are labelled and formatted appropriately

5. Study limitations need to be provided

6. PLOS authors have the option to publish the peer review history of their article (what does this mean?). If published, this will include your full peer review and any attached files.

Reviewer #1: No

Reviewer #2: No

---

## [Author Response · Author response to Decision Letter 0]

4 Oct 2022

Journal Requirements:

DONE

DONE

AGREED

4. Please ensure that you refer to Figures 3 and 4 in your text as, if accepted, production will need this reference to link the reader to the figure.

DONE

Reviewer #1: Thank you for your efforts. Please find the following comments for your consideration to enhance the paper:

Title: Finding what links? The purpose of the study highlighted in the abstract can be used to revise the title for clarity what the study is all about.

Dear reviewer,

Thank you for your comment.

The title has been revised to better represent the content of the study.

Abstract – Purpose, methods, findings and conclusion/implications; no need for some points will be discussed later.

Dear reviewer,

Thank you for your comment.

Correction has been done.

Introduction

• Employee can help… and can prevent… in what sense and are there references to support the claim?

• A common problem in a prison… Authors should be specific and discuss what they are sure about. They also need to provide evidence to support any arguments.

• When Indonesia faces … TL will be a tool or was considered an approach?

• Strong justifications for the study at the end of the introduction section will add value to the study.

Dear reviewer,

Thank you for your comment.

The first sentence in the Introduction has been deleted since it considers a claim.

The paragraph has given sufficient information and evidence regarding this overcapacity issue. However, the wording has been revised to deliver better understanding.

A statement has been added in the last paragraph of Introduction section.

Literature Review

An introductory sentence or expressions to set a stage for what is to be discussed in this section will be beneficial.

Transformational Leadership

The study describes or a study describes. Better still, in a study, TL is described as …

[7] describe… should be changed and cited appropriately.

Dear reviewer,

Thank you for your comment.

All the suggestions have been responded accordingly. Some additional sources have been added to improve the Literature Review (p. 4 and 5)

Hypotheses development

H1, H2 and H3 – It’s been established that TL has positive effects on job satisfaction and performance from the literature reviewed and discussed in the preceding sections. So, why the hypotheses again or do you want to explore it as you are not sure it can have positive effects in your context? The same goes for H4 and H5.

Is the figure showing the research methods or framework?

Dear reviewer,

Thank you for your comment.

Based on the pre-research interview, there was a phenomenon worth investigated. This underlined the establishment of hypotheses. The additional statement has been added in the last paragraph of Introduction section (p. 4)

The figure has been corrected to research framework.

Methods

Can be arranged as follows: research philosophy, approach, instrumentation, sample and sampling technique, data collection procedure, data analysis and ethical considerations

In the data analysis section – Authors should describe the methods employed to analyze the collected data. All the outcomes of the analyses should be presented in the results section. The tables and figures are far too many, though. Perhaps, the authors may want to present critical results helping them to address the goals and hypotheses of the study. Seven hypotheses may be considered too many for a study of this nature.

Dear reviewer,

Thank you for your comment.

We have added what we consider as research philosophy in the Method section. However, we decided not to reorder the arrangement as suggested since we consider that the present arrangement already includes all details necessary in methodology.

Some of the tables have been deleted and substituted with description in Data Analysis subsection.

Discussion should be robustly done using the extant literature and highlighting possible implications.

From the findings and conclusions, it is not clear what the study is contributing to the extant international literature. The authors may want to look at the novelty and unique contributions of the study.

The authors should also look at the in-text citations and fix typos/grammatical errors across the manuscript.

Dear reviewer,

Thank you for your comment.

More sources were included in the Discussion subsection.

More statements were added in the Conclusion section to add the novelty.

Reviewer #2: The authors of this manuscript have done a good job conducting SEM mediation analysis to assess their hypothesized model, which is appropriate from a logico-deductive standpoint. I think that overall this is an interesting and salient piece that can be appropriate for PLOS One, though I do have some areas of feedback to propose:

Dear reviewer,

Thank you for your comment.

1. Writing at times is not as clear as it could be - occasional grammar and spelling issues in particular

Dear reviewer,

Thank you for your comment.

The writing has been improved thoroughly.

2. I do not think the literature review has gone deep enough into core research, especially around the key theoretical pieces from which LMX and other leadership theories derived. I would suggest some additional work in this area - see the Northouse Leadership textbook for some helpful sources.

Dear reviewer,

Thank you for your comment.

Some sources have been added in Literature Review to strengthen the foundation including source from Northouse Leadership.

3. Methodology needs expansion, especially providing some detail on the context/setting (prison) and some additional information/literature support on the methodology

Dear reviewer,

Thank you for your comment.

Details of the setting have been added in the last paragraph of the Introduction section

4. Make sure tables and figures are labelled and formatted appropriately

Dear reviewer,

Thank you for your comment.

The suggestion has been responded accordingly.

5. Study limitations need to be provided

Dear reviewer,

Thank you for your comment.

Limitation of the study as well as future research suggestion have been added.

---

## [Editor Report · Decision Letter 1]

14 Oct 2022

Transformational Leaders' Approach to Overcapacity: A Study in Correctional Institutions

PONE-D-22-17395R1

Dear Dr. Anis Eliyana,

We’re pleased to inform you that your manuscript has been judged scientifically suitable for publication and will be formally accepted for publication once it meets all outstanding technical requirements.

Kind regards,

Rogis Baker, Ph.D

Academic Editor

PLOS ONE
---

## [Editor Report · Acceptance letter]

20 Oct 2022

PONE-D-22-17395R1 

Transformational Leaders' Approach to Overcapacity: A Study in Correctional Institutions 

Dear Dr. Eliyana:

I'm pleased to inform you that your manuscript has been deemed suitable for publication in PLOS ONE. Congratulations! Your manuscript is now with our production department. 

Kind regards, 

on behalf of

Dr. Rogis Baker 

Academic Editor

PLOS ONE